# Malondialdehyde, Antioxidant Defense System Components and Their Relationship with Anthropometric Measures and Lipid Metabolism Biomarkers in Apparently Healthy Women

**DOI:** 10.3390/biomedicines11092450

**Published:** 2023-09-03

**Authors:** Linas Černiauskas, Asta Mažeikienė, Eglė Mazgelytė, Eglė Petrylaitė, Aušra Linkevičiūtė-Dumčė, Neringa Burokienė, Dovilė Karčiauskaitė

**Affiliations:** 1Department of Physiology, Biochemistry, Microbiology and Laboratory Medicine, Institute of Biomedical Sciences, Faculty of Medicine, Vilnius University, M. K. Čiurlionio St. 21, LT-03101 Vilnius, Lithuania; asta.mazeikiene@mf.vu.lt (A.M.); egle.mazgelyte@mf.vu.lt (E.M.); ausra.linkeviciute@mf.vu.lt (A.L.-D.); dovile.karciauskaite@mf.vu.lt (D.K.); 2Clinics of Internal Diseases, Family Medicine and Oncology, Institute of Clinical Medicine, Faculty of Medicine, Vilnius University, LT-03101 Vilnius, Lithuania; neringa.burokiene@mf.vu.lt

**Keywords:** malondialdehyde, lipid peroxidation, antioxidants, oxidative stress, lipid metabolism markers

## Abstract

Cardiovascular diseases are the leading cause of mortality worldwide. Since atherosclerosis, an inflammatory, lipid-driven disease, is an underlying basis for the development of cardiovascular disease, it is important to understand its relationship with confounding factors, such as oxidative lipid degradation. In contrast, circulating antioxidants prevent oxidative lipid damage, and therefore, may be associated with reduced development of atherosclerosis. We aimed to assess oxidative lipid degradation biomarker malondialdehyde (MDA) and antioxidant defense system components, total antioxidant capacity (TAC) and superoxide dismutase (SOD) inhibition rate levels, in healthy women and evaluate their relationships with age, anthropometric measures, and lipid metabolism biomarkers. The study included 86 healthy middle-aged women. MDA in human serum samples was evaluated by HPLC, and the TAC and SOD inhibition rates were measured by photometric methods. MDA was found to be associated with age, total cholesterol, non-HDL cholesterol, apolipoprotein B and triacylglycerols. TAC was shown to be associated with age, BMI, and waist circumference, as well as lipid metabolism biomarkers apolipoprotein B and triacylglycerol, while SOD inhibition rate was only associated with total cholesterol, apolipoprotein B and triacylglycerols. In conclusion, the association of oxidative status indices, MDA, TAC and SOD, with cardiovascular risk factors suggests that they could be additional useful biomarkers in the research of aging, obesity, and atherosclerosis pathogenesis.

## 1. Introduction

Cardiovascular diseases (CVDs) are the leading cause of mortality worldwide, resulting in 20.5 million deaths in 2021 [1]. Atherosclerosis is the main underlying cause of CVDs and is defined as lipid-driven chronic inflammatory disease, which has a detrimental effect on arteries [2,3,4]. Pathological events, such as endothelial dysfunction, transendothelial LDL particle infiltration and their lipid cargo deposition in intima, immune system-mediated inflammation, proliferation of vascular smooth muscle cells and changes in the extracellular matrix are the key players in the development of atherosclerosis [3,4,5,6,7]. Inflammation in atherosclerosis is strongly associated with free radicals and reactive oxygen species (ROS), which have deleterious effects on lipoproteins and their lipid cargo [8,9,10]. Lipid peroxidation occurs when polyunsaturated fatty acids (PUFAs) in lipoproteins and their lipid cargo are attacked by free radicals and ROS. The primary products of lipid peroxidation are unstable peroxyl radicals and lipid hydroperoxides. Subsequently, these unstable products turn into stable secondary products of oxidative lipid degradation, such as malondialdehyde (MDA) [10,11,12]. MDA is able to modify proteins, nucleotides, phospholipids, and free amino acids by forming adducts, which are linked to exacerbated immune response and apoptosis [13]. To combat deleterious effects of free radicals and ROS there are two main antioxidant systems: enzymatic and nonenzymatic. Enzymatic antioxidant system consists of variety of antioxidant enzymes that neutralize free radicals and ROS, such as glutathione reductase (GR), superoxide dismutase (SOD) and catalase (CAT). The nonenzymatic antioxidant system consists of molecules that act as electron donors and can neutralize free radicals and ROS. Examples of this system are endogenous and exogenous molecules: glutathione (GSH), ascorbic acid, tocopherols, carotenoids, and uric acid [7,14,15].

The aim of this study was to determine the stable secondary lipid peroxidation product MDA, the enzymatic antioxidative system component SOD inhibition rate and the summarized measure of both antioxidant defense systems total antioxidant capacity (TAC) levels in human serum and their relations with anthropometric measures and lipid metabolism biomarkers.

## 2. Materials and Methods

### 2.1. Study Design

The design of this cross-sectional pilot study is depicted in Figure 1.

### 2.2. Study Participants

This cross-sectional study included 86 apparently healthy women (aged 50–64) participating in the national cardiovascular disease prevention program. Women with acute or chronic diseases, diabetes mellitus or chronic kidney disease were excluded from the study. All participants provided written informed consent prior to the study. The study protocol was approved by the Vilnius Regional Biomedical Research Ethics Committee (No. 2020/8-1254-735). The procedures of this study comply with the tenets of the Declaration of Helsinki.

### 2.3. Anthropometric Parameters and Preparation of Blood Samples

Anthropometric parameters (body mass index (BMI), waist circumference (WC)) were assessed by trained personnel. Cut-off values for BMI and WC were chosen according to commonly accepted reference ranges for women, 25.0 kg/m^2^ for BMI and 88.0 cm for WC. In addition, venous blood samples were collected after overnight fast, allowed to clot for 30 min. at room temperature and centrifuged for 10 min. at 2000× *g*. Separated blood serum was aliquoted, frozen and stored at −80 °C. Prior to analysis, serum aliquots were thawed at room temperature and thoroughly mixed.

### 2.4. Lipid Metabolism Biomarker Analysis

Lipid metabolism biomarkers (total cholesterol, high-density lipoprotein cholesterol (HDL cholesterol), triacylglycerols) in blood serum were measured by enzymatic methods (Architect ci8200, Abbott, Chicago, IL, USA). Low-density lipoprotein cholesterol (LDL cholesterol) was calculated using Friedewald’s equation and non-HDL cholesterol was calculated by subtracting HDL-cholesterol concentration from total-cholesterol concentration. Furthermore, apolipoprotein measurements (Apolipoprotein AI, Apolipoprotein B) were performed by immunonephelometric methods using routine analytical methods (Siemens BN II, Siemens Medical Solutions, Malvern, PA, USA). Dyslipidemia status was confirmed if one or more lipid metabolism parameter abnormality was observed: total cholesterol > 5.2 mmol/L, LDL cholesterol > 2.58 mmol/L, HDL cholesterol < 1.03 mmol/L or triacylglycerols > 1.7 mmol/L.

### 2.5. Serum Antioxidant and Oxidative Stress Biomarker Analysis

Serum total antioxidant capacity (TAC) was evaluated using antioxidant assay kit (Antioxidant Assay Kit, catalog number MAK334, Sigma-Aldrich, St. Louis, MO, USA) according to the instructions provided by the manufacturer. In short, total antioxidant capacity assay is based on Cu^2+^ ion reduction by the antioxidants in the sample to Cu^+^. Subsequently, Cu^+^ reacts with dye reagent and forms colored compound, which absorbs light at 570 nm. The absorbance is directly proportional to the total antioxidant concentration in the sample. Total antioxidant capacity is calculated as Trolox equivalents in µmol/L.

Serum superoxide dismutase (SOD) inhibition rate (%) was determined by SOD determination kit (SOD determination kit, catalog number 19160, Sigma-Aldrich, Darmstadt, DE, USA) according to the instructions provided by the manufacturer. Summarizing the method, the product of enzymatic reaction of xanthine oxidase (XO) and tetrazolium salt is colored formazan dye. The reaction is inhibited by SOD in the sample and the inhibition rate is determined by the decrease in the color development at 440 nm. SOD inhibition rate is reported in %.

Malondialdehyde concentration in blood serum was determined according to our previously published methodology by Mazgelytė et al. [16]. MDA concentration is reported in µmol/L.

### 2.6. Statistical Analysis

Statistical analysis was carried out using the R software (version 4.3.0). To assess the normality of variables, the Shapiro–Wilk test was used. Quantitative variables are presented as mean ± standard deviation (SD) for normally distributed, and median (interquartile range) (IQR) for non-normally distributed variables. To compare the median values of anthropometric and lipid metabolism biomarkers in the groups based on MDA, TAC and SOD inhibition rate median values, a nonparametric Kruskal–Wallis test was performed. Also, the median values of MDA, TAC and SOD inhibition rate in BMI, WC, normolipidemic and dyslipidemia groups were compared by performing a nonparametric Kruskal–Wallis test. To quantify the strength of the correlation between biomarkers the Spearman’s rank coefficient was used. The level of statistical significance was set at 0.05 for two-tailed testing.

## 3. Results

### 3.1. Study Group Characteristics

The descriptive statistics of lipid metabolism, antioxidative system and oxidative stress biomarkers, as well as anthropometric parameters, are depicted in Table 1.

### 3.2. Age, Anthropometric Parameters, Antioxidative Defense System Biomarkers and MDA

The results showed that women in higher MDA concentration group (lower than the median value) were older. No statistically significant differences were observed in BMI and waist circumference between women in different MDA concentration groups (Table 2).

Furthermore, the analysis revealed that women in the higher TAC concentration group had higher BMI and waist circumference values (Table 3).

Correlation analysis revealed a statistically significant although weak relationship between serum MDA values and age. Moreover, TAC had a statistically significant weak relationship with age, but moderate strength relationships with BMI and waist circumference. There were no statistically significant relationships between serum MDA values and BMI or waist circumference. Finally, there were no statistically significant relationships between SOD inhibition rate and age, BMI, or waist circumference (Table 4).

### 3.3. Lipid Metabolism Biomarkers, Antioxidative Defense System Biomarkers and MDA

The analysis of lipid metabolism biomarkers showed that women in the higher MDA concentration group (equal to or higher than the median value) had higher total cholesterol values. Interestingly, a borderline statistically significant result was observed in the comparison of ApoB levels between the groups: higher ApoB concentrations were observed in women in the group with higher MDA concentration. No statistically significant differences were observed in ApoAI, ApoB, HDL cholesterol, LDL cholesterol, non-HDL cholesterol, and triacylglycerol values between women in different MDA concentration groups (Table 5).

In addition, women in higher TAC concentration group had higher ApoB and triacylglycerol levels. The same observation about ApoB levels was made in women group with higher SOD inhibition rate (equal to or higher than the median value). No statistically significant differences between women in TAC or SOD inhibition rate value groups were observed between ApoAI, total cholesterol, HDL cholesterol, LDL cholesterol, and non-HDL cholesterol values. Lastly, there were no statistically significant differences between triacylglycerol levels for women in different SOD inhibition rate groups (Table 6).

Correlation analysis revealed that there were statistically significant weak relationships between MDA and SOD inhibition rate levels and triacylglycerol concentration. Interestingly, TAC and triacylglycerol concentration had a statistically significant moderate strength relationship. In addition, MDA, TAC and SOD inhibition rate values had statistically significant but weak relationships with ApoB concentration. Furthermore, MDA and SOD inhibition rate values had statistically significant weak relationships with total-cholesterol concentration. There was a statistically significant relationship between MDA and non-HDL cholesterol values, although the strength of relationship was weak. Lastly, there were no other statistically significant relationships observed between lipid metabolism biomarkers and MDA, TAC or SOD inhibition rate (Table 7).

### 3.4. Lipid Peroxidation and Antioxidant Defense System Biomarkers and BMI, Waist Circumference and Dyslipidemia

The analysis revealed that women with higher BMI or waist circumference values had higher TAC median values. Furthermore, no statistically significant differences were observed between MDA and SOD inhibition rate median values in different BMI and waist circumference groups (Table 8).

The analysis of MDA median values in normolipidemic and dyslipidemia groups revealed that women in dyslipidemia group had higher MDA median values than women in normolipidemic group. In contrast, no statistically significant differences between TAC and SOD inhibition rate median values in normolipidemic and dyslipidemia groups were observed (Table 9).

## 4. Discussion

Investigation of antioxidative system components and MDA and their relationship with age and anthropometric parameters revealed that MDA and TAC concentrations were positively associated with age. Moreover, TAC concentration was found to be elevated in higher BMI and waist circumference women groups and correlation analysis confirmed this finding, as TAC was positively associated with BMI and waist circumference. The positive relationship between MDA and age observed in this study is in concordance with the results of other studies that observedan MDA increase with age [17,18,19]. The age-related increase of MDA in the human body could be explained by the oxidative stress theory of aging. The main rationale behind this theory is that the accumulation of oxidative damage to macromolecules, such as lipids, nucleic acids and proteins, lead to cell senescence and reduced capacity to neutralize free radicals and ROS [20,21]. In this study, the positive relationship between TAC and age, BMI, and waist circumference values were observed. Limberaki et al. concluded that middle-aged people (36–60 years) had the highest TAC serum values compared to young (16–35 years) and elderly (60–90) age groups but did not carry out correlation analysis between TAC and age values [22]. The positive relationship between TAC values and age might be explained by the cellular response against oxidative stress via nuclear factor erythroid 2-related factor 2 (Nrf2). Nrf2 regulates the expression of genes that encode enzymes involved in ROS and free radical detoxification [23]. Subsequently, the increased cellular synthesis of these enzymes may positively affect levels of circulating antioxidants in human blood. Similar observations were made by Gawron-Skarbek A. et al. [24] who concluded that TAC values were positively associated with BMI and waist circumference in healthy men. In a cross-sectional study of Iranian women by Mozaffari et al., TAC was evaluated by three different methodologies of which Trolox equivalent antioxidant capacity (TEAC) was similar to the method described in this study. The study concluded that there was a statistically significant association between TEAC and BMI values but there was no statistically significant association between TEAC values and waist circumference [25]. The positive relationship between TAC and BMI, as well as waist circumference, may be attributed to higher overall concentrations of all biologically active molecules, including antioxidants. Notably, there is no current research reporting human body composition and antioxidant distribution or retention, which may help explain the results found in this and other studies. Finally, it is important to note that there are few studies exploring the relationship between TAC, age and anthropometric parameters with similar methodologies employed in this study.

The analysis of antioxidative defense system components, MDA and lipid metabolism biomarkers revealed that ApoB and triacylglycerol concentrations were positively associated with MDA, TAC and SOD inhibition rate values. Furthermore, total-cholesterol concentration was positively associated with MDA and SOD inhibition rate levels and non-HDL-cholesterol concentration was positively associated with MDA levels. In addition, MDA concentration was showed to be increased in women with dyslipidemia compared to normolipidemic women group. The relationships between MDA, total cholesterol and triacylglycerols were explored in study by Dziegielewska-Gesiak et al. [26]. In concordance with our findings, they concluded that MDA was positively associated with total cholesterol and triacylglycerol levels in a sample consisting of middle-aged (39–58 years) and elderly (67–74 years) people. Contrastingly, they found that MDA was negatively associated with HDL-cholesterol concentrations, which was not observed in our study. Furthermore, in the case-control study (healthy people vs. people with asymptomatic cholelithiasis) by Atamer et al., the positive associations between MDA and total cholesterol and triacylglycerols were observed only in asymptomatic cholelithiasis people group [27]. Furthermore, in the same study there was no statistically significant association between HDL-cholesterol concentration and MDA values in healthy people group, which is in concordance with our findings. The positive association between total cholesterol and triacylglycerol and MDA, as well as higher MDA levels in dyslipidemia, are expected findings, as MDA is a product of oxidative lipid degradation, thus increased levels of lipids in serum may result in increased levels of MDA. In concordance with our observations, neither Dziegielewska-Gesiak et al. nor Atamer et al. found statistically significant associations between MDA and LDL cholesterol [26,27]. Even though LDL cholesterol is a surrogate ApoB biomarker, we found that ApoB but not LDL cholesterol was positively associated with MDA concentration. The possible theoretical link between MDA and ApoB concentrations may be MDA’s ability to form adducts with proteins [13]. Theoretically, MDA-modified ApoB may contribute to total MDA concentration and thus result in positive association between these two biomarkers. Further studies of MDA-modified apolipoproteins should be carried out to confirm or reject this hypothesis, as we observed no positive association between MDA and ApoAI levels. Dziegielewska-Gesiak et al. found negative association between total antioxidant status (TAS) and HDL-cholesterol concentration but did not report any statistically significant associations between TAS and other lipid metabolism biomarkers [26]. We observed the same negative association between these two parameters, but the result was not statistically significant. As there is a small number of studies exploring relationships between antioxidative system components, lipid metabolism biomarkers and their abnormalities, such as dyslipidemia, further research is needed to determine the relationship between these biomarker groups. This was a pilot cross-sectional study that observed antioxidant defense system and MDA relationship with age, anthropometric measures, and lipid metabolism biomarkers at one timepoint. Although the results showed statistically significant associations between malondialdehyde, antioxidant defense system components and lipid metabolism biomarkers, the application of these measures for the assessment of CVD risk is debatable. Currently, CVD risk estimation, individual-level interventions and treatment goals are based on the assessment of conventional risk factors including low physical activity, unbalanced diet, obesity, smoking, dyslipidemia, hypertension, and the presence of diabetes [28]. It is suggested that atherosclerosis and coronary artery calcification may be the result of oxidative stress and reactive oxygen species are involved in endothelial dysfunction, atheromatous plaque formation and rupture, and measurement of circulating biomarkers of oxidative stress in serum or plasma is challenging because of the highly reactive nature of these molecules [28,29,30]. Moreover, there is no consensus regarding which oxidative stress/antioxidant defense system measure (e.g., plasma total antioxidant status, F2-isoprostanes, lipid peroxides, thiobarbituric acid-reactive substances and monoaldehyde, biomarkers of protein peroxidation) is the most reliable “gold” standard method for the evaluation of oxidative damage in vivo [30]. Thus, larger prospective studies are needed to identify a universal and “easy to measure” marker of oxidative stress, which might be involved in the current CVD prevention guidelines. Additionally, there are varying methodologies for MDA, TAC and SOD inhibition rate determination in human blood samples presented in the scientific literature. Currently, there is a lack of standardization for these methods, resulting in poorly comparable and highly variable results between different studies. Furthermore, the findings of the study are applicable only to a population of middle-aged women and cannot be applied to other study groups. For example, women are less susceptible to oxidative stress and there are clear differences in the expression and/or activities of antioxidant enzymes (catalase, superoxide dismutase, glutathione peroxidase) between males and females [31]. Finally, there were no data available on menopausal status of study participants. It is well-established that menopause is associated with changes in lipid metabolism biomarkers such as lipoproteins, LDL cholesterol, HDL cholesterol and triacylglycerols [32]. Therefore, we would like to disclose that these changes were not addressed in this study, and we encourage other researchers to include the menopausal status of participants in similar studies.

## 5. Conclusions

In conclusion, the association of oxidative status indices, MDA, TAC and SOD inhibition rates, with cardiovascular risk factors suggests that they could be additional useful biomarkers in the research of aging, obesity, and atherosclerosis pathogenesis.

Although our results showed weak or moderate statistically significant associations, it is important to note that there are numerous methods for malondialdehyde and antioxidant defense system component determination, which results in high variability and poor comparability of the studies. Lastly, as this was a pilot study, we would like to note that current results and their statistical significance would need to be confirmed in a study with a higher number of participants.

## Figures and Tables

**Figure 1 biomedicines-11-02450-f001:**
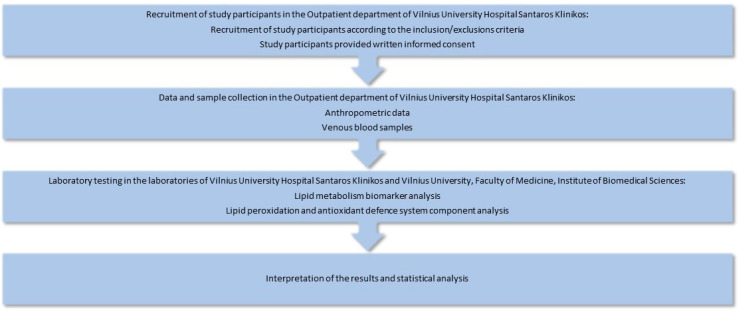
The design of the study.

**Table 1 biomedicines-11-02450-t001:** Anthropometric, lipid metabolism and antioxidative defense system component characteristics of the study participants.

Variable (n = 86)	Mean ± SD or Median (IQR)	Range
Anthropometric parameters		
Age (years)	55 (7)	50–64
BMI * (kg/m^2^)	27.98 ± 5.45	17.97–43.44
Waist circumference (cm)	87.86 ± 12.23	62.00–127.00
Lipid metabolism biomarkers		
ApoAI * (g/L)	1.54 (0.25)	0.86–2.49
ApoB * (g/L)	0.91 (0.35)	0.09–1.61
Total cholesterol (mmol/L)	5.80 (1.41)	4.13–10.69
HDL cholesterol * (mmol/L)	1.59 (0.62)	0.85–2.90
LDL cholesterol * (mmol/L)	3.47 (1.07)	1.95–8.88
Non-HDL cholesterol * (mmol/L)	4.01 (1.34)	2.78–9.47
Triacylglycerols (mmol/L)	1.15 (0.82)	0.53–3.54
Antioxidative system and oxidative stress biomarkers		
TAC * (µmol/L)	394.6 (112.22)	154.20–1187.50
SOD inhibition rate * (%)	76.08 (8.26)	53.24–89.22
Malondialdehyde (µmol/L)	1.72 (0.55)	0.77–3.78

* ApoAI—apolipoprotein AI, ApoB—apolipoprotein B, BMI—body mass index, HDL cholesterol—high-density lipoprotein cholesterol, LDL cholesterol—low-density lipoprotein cholesterol, non-HDL cholesterol—non-high-density lipoprotein cholesterol, SOD inhibition rate—superoxide dismutase inhibition rate, TAC—total antioxidant capacity.

**Table 2 biomedicines-11-02450-t002:** Age and anthropometric measure median values in different MDA value groups.

Parameter	MDA * < 1.72 (µmol/L)	MDA * ≥ 1.72 (µmol/L)	*p*-Value
Age (years)	54 (6.5)	56 (7.0)	0.014
BMI * (kg/m^2^)	26.70 (7.64)	27.14 (7.88)	0.832
Waist circumference (cm)	85 (14.5)	86 (18.5)	0.873

* BMI—body mass index, MDA—malondialdehyde.

**Table 3 biomedicines-11-02450-t003:** Age and anthropometric measure median values in different TAC and SOD inhibition rate value groups.

Parameter	TAC *< 394.6 (µmol/L)	TAC * ≥ 394.6 (µmol/L)	*p*-Value	SOD Inhibition Rate * < 76.08 (%)	SOD Inhibition Rate * ≥ 76.08 (%)	*p*-Value
Age (years)	54 (6.0)	56 (7.0)	0.205	54 (7.0)	55 (7.0)	0.558
BMI * (kg/m^2^)	24.34 (4.05)	29.76 (6.94)	3.384 × 10^−5^	26.95 (7.80)	27.14 (8.36)	0.816
Waist circumference (cm)	80 (10.5)	93 (14.0)	3.539 × 10^−5^	89 (16.5)	85 (15.5)	0.551

* BMI—body mass index, SOD inhibition rate—superoxide dismutase inhibition rate, TAC—total antioxidant capacity.

**Table 4 biomedicines-11-02450-t004:** Correlations between age, anthropometric measures, lipid peroxidation and antioxidant defense system biomarkers.

Parameter	MDA * (µmol/L)	TAC * (µmol/L)	SOD Inhibition Rate * (%)
Age (years)	R = 0.28, *p* = 0.009	R = 0.22, *p* = 0.039	R = 0.03, *p* = 0.759
BMI * (kg/m^2^)	R = 0.06, *p* = 0.581	R = 0.43, *p* = 3.319 × 10^−5^	R = −0.002, *p* = 0.988
Waist circumference (cm)	R = 0.05, *p* = 0.587	R = 0.42, *p* = 6.68 × 10^−5^	R = −0.08, *p* = 0.456

* BMI—body mass index, MDA—malondialdehyde, SOD inhibition rate—superoxide dismutase inhibition rate, TAC—total antioxidant capacity.

**Table 5 biomedicines-11-02450-t005:** Lipid metabolism biomarker values in different MDA value groups.

Parameter	MDA * < 1.72 (µmol/L)	MDA * ≥ 1.72 (µmol/L)	*p*-Value
ApoAI * (g/L)	1.51 (0.26)	1.54 (0.33)	0.138
ApoB * (g/L)	0.87 (0.27)	0.93 (0.37)	0.049
Total cholesterol (mmol/L)	5.26 (1.36)	6.18 (1.14)	0.007
HDL cholesterol * (mmol/L)	1.53 (0.56)	1.63 (0.63)	0.118
LDL cholesterol * (mmol/L)	3.33 (1.25)	3.56 (0.92)	0.194
Non-HDL cholesterol * (mmol/L)	3.84 (1.34)	4.14 (1.53)	0.077
Triacylglycerols (mmol/L)	1.03 (0.65)	1.23 (0.81)	0.136

* ApoAI—apolipoprotein AI, ApoB—apolipoprotein B, HDL cholesterol—high-density lipoprotein cholesterol, LDL cholesterol—low-density lipoprotein cholesterol, MDA—malondialdehyde, non-HDL cholesterol—non-high-density lipoprotein cholesterol.

**Table 6 biomedicines-11-02450-t006:** Lipid metabolism biomarker values in different TAC and SOD inhibition rate value groups.

Parameter	TAC * < 394.6 (µmol/L)	TAC * ≥ 394.6 (µmol/L)	*p*-Value	SOD Inhibition Rate * < 76.08 (%)	SOD Inhibition Rate * ≥ 76.08 (%)	*p*-Value
ApoAI * (g/L)	1.54 (0.33)	1.50 (0.22)	0.720	1.54 (0.23)	1.53 (0.34)	0.666
ApoB * (g/L)	0.85 (0.17)	0.97 (0.37)	0.012	0.85 (0.32)	0.95 (0.32)	0.034
Total cholesterol (mmol/L)	5.62 (1.27)	6.01 (1.46)	0.364	5.46 (1.30)	6.01 (1.23)	0.110
HDL cholesterol * (mmol/L)	1.64 (0.64)	1.53 (0.54)	0.245	1.56 (0.49)	1.61 (0.68)	0.601
LDL cholesterol * (mmol/L)	3.45 (0.87)	3.56 (1.17)	0.966	3.25 (1.13)	3.61 (0.92)	0.203
Non-HDL cholesterol * (mmol/L)	3.98 (0.96)	4.27 (1.60)	0.238	3.88 (1.42)	4.27 (1.26)	0.108
Triacylglycerols (mmol/L)	0.96 (0.45)	1.58 (0.92)	1.671 × 10^−6^	1.11 (0.50)	1.28 (1.06)	0.146

* ApoAI—apolipoprotein AI, ApoB—apolipoprotein B, HDL cholesterol—high-density lipoprotein cholesterol, LDL cholesterol—low-density lipoprotein cholesterol, non-HDL cholesterol—non-high-density lipoprotein cholesterol, SOD inhibition rate—superoxide dismutase inhibition rate, TAC—total antioxidant capacity.

**Table 7 biomedicines-11-02450-t007:** Correlations between lipid metabolism, lipid peroxidation and antioxidant defense system biomarkers.

Parameter	MDA * (µmol/L)	TAC * (µmol/L)	SOD Inhibition Rate * (%)
ApoAI * (g/L)	R = 0.17, *p* = 0.111	R = 0.02, *p* = 0.869	R = −0.02, *p* = 0.859
ApoB * (g/L)	R = 0.32, *p* = 0.003	R = 0.37, *p* = 5.085 *10^−4^	R = 0.22, *p* = 0.034
Total cholesterol (mmol/L)	R = 0.35, *p* = 0.001	R = 0.20, *p* = 0.063	R = 0.22, *p* = 0.034
HDL cholesterol * (mmol/L)	R = 0.12, *p* = 0.288	R = −0.09, *p* = 0.397	R = −0.07, *p* = 0.498
LDL cholesterol * (mmol/L)	R = 0.20, *p* = 0.063	R = 0.08, *p* = 0.454	R = 0.13, *p* = 0.221
Non-HDL cholesterol * (mmol/L)	R = 0.26, *p* = 0.016	R = 0.21, *p* = 0.051	R = 0.21, *p* = 0.055
Triacylglycerols (mmol/L)	R = 0.24, *p* = 0.024	R = 0.54, *p* = 7.289 *10^−8^	R = 0.22, *p* = 0.034

* ApoAI—apolipoprotein AI, ApoB—apolipoprotein B, HDL cholesterol—high-density lipoprotein cholesterol, LDL cholesterol—low-density lipoprotein cholesterol, MDA—malondialdehyde, non-HDL cholesterol—non-high-density lipoprotein cholesterol, SOD inhibition rate—superoxide dismutase inhibition rate, TAC—total antioxidant capacity.

**Table 8 biomedicines-11-02450-t008:** Lipid peroxidation and antioxidant defense system biomarkers in different BMI and WC groups.

Parameter	BMI * < 25.0 (kg/m^2^)	BMI * ≥ 25.0 (kg/m^2^)	*p*-Value	WC * < 88.0 (cm)	WC * ≥ 88.0 (cm)	*p*-Value
N	32	54	-	45	51	-
MDA * (µmol/L)	1.73 (0.62)	1.70 (0.51)	0.968	1.71 (0.51)	1.73 (0.54)	0.202
TAC * (µmol/L)	337.84 (87.19)	421.92 (92.77)	4.038 × 10^−5^	349.65 (98.93)	422.15 (108.75)	1.313 × 10^−4^
SOD inhibition rate * (%)	76.33 (0.87)	76.08 (8.28)	1.000	76.87 (8.21)	75.08 (9.62)	0.492

* BMI—body mass index, MDA—malondialdehyde, SOD inhibition rate—superoxide dismutase inhibition rate, TAC—total antioxidant capacity, WC—waist circumference.

**Table 9 biomedicines-11-02450-t009:** Lipid peroxidation and antioxidant defense system biomarkers in normolipidemic and dyslipidemia groups.

Parameter	Normolipidemic Group	Dyslipidemia Group	*p*-Value
N	32	54	-
MDA * (µmol/L)	1.58 (0.41)	1.79 (0.60)	3.946 × 10^−2^
TAC * (µmol/L)	386.97 (107.08)	402.14 (116.38)	0.911
SOD inhibition rate * (%)	75.85 (5.60)	76.84 (9.23)	0.514

* MDA—malondialdehyde, SOD inhibition rate—superoxide dismutase inhibition rate, TAC—total antioxidant capacity, WC—waist circumference.

## Data Availability

The data supporting the reported results are archived in the National Open Access Research Data Archive (MIDAS) at www.midas.lt. (accessed on 24 October 2021).

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
