# Peer review of "Malondialdehyde, Antioxidant Defense System Components and Their Relationship with Anthropometric Measures and Lipid Metabolism Biomarkers in Apparently Healthy Women"

_biomedicines, 2023, doi:10.3390/biomedicines11092450_

Round 1
Reviewer 1 Report
The manuscript “Malondialdehyde, antioxidant defence system components and their relation with anthropometric measures and lipid metabolism biomarkers in apparently healthy women” by Linas Černiauskas et al. The manuscript is written in standard English; however, it has several grammatical and typographical errors. After thoroughly reviewing it, I feel the manuscript needs substantial revision as the data presented in the manuscript is not up to mark for this journal.
Comments:
1. In the abstract section, I suggest better rewriting the conclusion.
2. I will suggest citing the latest literature in the introduction section and rewriting it correctly.
3. I will suggest adding experimental design as a Schematic diagram
4. The authors should carefully check grammatical errors.
5. I will suggest explaining the discussion section properly.
6. Authors have shown the relation between Oxidative lipid degradation biomarker MDA and antioxidant defence system components as valuable biomarkers in assessing atherosclerosis risk. However, the ultimate cause of atherosclerosis is chronic inflammation. However, the authors have not analysed any inflammatory markers. Inflammatory markers must be analyzed.
Author Response
Dear Editor,
We thank the Reviewers for the valuable comments. They are given below in black fonts together with our answers:
- In the abstract section, suggest better rewriting the conclusion.
As suggested, we have reformulated the conclusion in the abstract:
In conclusion, the association of oxidative status indices, MDA, TAC and SOD, with cardiovascular risk factors suggests that they could be additional useful biomarkers in the research of aging, obesity, and atherosclerosis pathogenesis.
- I will suggest citing the latest literature in the introduction section and
rewriting it correctly.
We have carefully revised the Introduction section of our manuscript and added the newest available epidemiological data and reference. However, we believe that some older references should remain in the manuscript as they refer to original ideas and understanding of oxidative stress and atherosclerosis pathogenesis.
- I will suggest adding experimental design as a Schematic diagram
A schematic diagram depicting experimental design was added to the manuscript.
4 The authors should carefully check grammatical errors.
The manuscript was carefully checked by native English-speaking professional.
- will suggest explaining the discussion section properly.
Thank you for the suggestion. The discussion section was revised.
- Authors have shown the relation between Oxidative lipid degradation biomarker MDA and antioxidant defence system components as valuable biomarkers in assessing atherosclerosis risk. However, the ultimate cause of atherosclerosis is chronic inflammation. However, the authors have not analysed any inflammatory markers. Inflammatory markers must he analyzed.
Thank you for the remark. We totally agree that inflammation plays a crucial role in atherosclerosis pathogenesis, therefore measurement of C-reactive protein, as an indicator of inflammatory status, is included in many cardiovascular prevention programs. However, the goal of this study was to evaluate measurable indices of oxidative status and their association with lipid biomarkers. Our results suggests that these simple measurable markers of oxidative stress could be of value in further research of atherosclerosis mechanism, including their role in promoting and maintaining inflammatory process.
Reviewer 2 Report
The submitted article aimed to assess the relationships between serum biomarkers of oxidative stress and antioxidant capacity and anthropometric measures and lipid biomarkers. The authors conclude that components of oxidative-antioxidative system could be applied as useful biomarkers in assessing the risk of atherosclerosis.
Major comments:
1.There is a serious flaw in the study design. The study group is very heterogenous and includes subjects with severe obesity (BMI>44kg/m2; high waist circumference) and subjects with dyslipidemia and also normal weight and probably normolipidemia. The broad range of levels of TAC and MDA reflect also the group heterogeneity. Both, obesity and hyperlipidemia are established risk factors of cardiovascular diseases. In my opinion, for the purpose of statistical analysis the study group should be divided at least into subgroups with normal body weight and overweight/obesity and/or subgroups with normolipidemia and hyperlipidemia, applying the generally accepted cut-offs for TC, TG and HDL-C and then compared. Such approach would make sense. If the study aimed to assess the relationships of ox/anti-ox biomarkers with atherosclerosis risk – subjects with obesity and/or dyslipidemia are at higher risk thus the relationships in this subgroup would be informative. Moreover, there is no data on menopausal status of women (some of them may be perimenopausal), which is of great importance with regard to lipid metabolism, in particular hypeetriglyceridemia.
2.Levels of lipid biomarkers do not differ significantly between groups divided at cut-off accepted for MDA, except total cholesterol; for apoB there is only borderline difference. At cut-offs applied for TAC and SOD significant differences were found for apoB whereas for TG only at a cut-off for TAC.
3.Correlations between MDA, TAC and age are very weak; more interesting is a good significant correlation found between TAC and WC. Correlations between MDA, TAC and SOD with apoB and TG, though significant, are weak or very weak, except this of TG with TAC. Correlations do not say much and sometimes may be quite accidental.
4.Methods for measuring MDA, TAC and SOD are not standardized and not harmonized as the authors admit themselves; moreover these are time-consuming methods which could be hardly applied in clinical and laboratory practice. Evaluation of new potential biomarkers should clearly indicate their advantage when applied in the diagnosis. Finally, based on presented data, I do not see any additional value of measuring MDA, TAC and SOD as biomarkers of atherosclerosis risk.
Minor comments:
1.The terms “apoproteins A or B” are wrongly used, instead apolipoproteins A or B should be used.
2. Surprisingly, section 3- Results (lines 120-122) starts with something like instruction “how to prepare the manuscript”.
English is fine, minor corrections may be introduced.
Author Response
Dear Editor,
We thank the Reviewers for the valuable comments. They are given below together with our answers to the comments:
Reviewer 2:
1.There is a serious flaw in the study design. The study group is very heterogenous and includes subjects with severe obesity (BMI>44kg/m2; high waist circumference) and subjects with dyslipidemia and also normal weight and probably normolipidemia. The broad range of levels of TAC and MDA reflect also the group heterogeneity. Both, obesity and hyperlipidemia are established risk factors of cardiovascular diseases. In my opinion, for the purpose of statistical analysis the study group should be divided at least into subgroups with normal body weight and overweight/obesity and/or subgroups with normolipidemia and hyperlipidemia, applying the generally accepted cut-offs for TC, TG and HDL-C and then compared. Such approach would make sense. If the study aimed to assess the relationships of ox/anti-ox biomarkers with atherosclerosis risk – subjects with obesity and/or dyslipidemia are at higher risk thus the relationships in this subgroup would be informative. Moreover, there is no data on menopausal status of women (some of them may be perimenopausal), which is of great importance with regard to lipid metabolism, in particular hypeetriglyceridemia.
Thank you for the remark. We supplemented our publication with additional statistical analysis of MDA, TAC and SOD inhibition rate indices in different BMI and waist circumference groups where clinically accepted cut off values for these parameters were used to differentiate between subjects with normal weight and overweight/obesity status. Results subsection 3.4. The analysis revealed that only TAC had statistically significant results.
3.4. Lipid peroxidation and antioxidant defense system biomarkers and BMI, waist circumference and dyslipidemia
The analysis revealed that women with higher BMI or waist circumference values had higher TAC median values. Furthermore, no statistically significant differences were observed between MDA and SOD inhibition rate median values in different BMI and waist circumference groups (Table 8).
Table 8. Lipid peroxidation and antioxidant defense system biomarkers in different BMI and WC groups.
|
Parameter |
BMI* < 25.0 (kg/m2) |
BMI* ≥ 25.0 (kg/m2) |
p-value |
WC* < 88.0 (cm) |
WC* ≥ 88.0 (cm) |
p-value |
|
N |
32 |
54 |
- |
45 |
51 |
- |
|
MDA* (µmol/l)
|
1.73 (0.62) |
1.70 (0.51) |
0.968 |
1.71 (0.51) |
1.73 (0.54) |
0.202 |
|
TAC* (µmol/l)
|
337.84 (87.19) |
421.92 (92.77) |
4.038*10-5 |
349.65 (98.93) |
422.15 (108.75) |
1.313*10-4 |
|
SOD inhibition rate* (%)
|
76.33 (0.87) |
76.08 (8.28) |
1.000 |
76.87 (8.21) |
75.08 (9.62) |
0.492 |
*BMI – body mass index, MDA – malondialdehyde, SOD inhibition rate – superoxide dismutase inhibiton rate, TAC – total antioxidant capacity, WC – waist circumference.
Furthermore, we also performed analysis on MDA, TAC and SOD inhibition rate between normolipidemic and dyslipidemia group, which revealed that MDA median values were higher in dyslipidemia group. The results of this analysis were added to results subsection 3.4.
The analysis of MDA median values in normolipidemic and dyslipidemia groups revealed that women in dyslipidemia group had higher MDA median values than women in normolipidemic group. In contrast, no statistically significant differences between TAC and SOD inhibition rate median values in normolipidemic and dyslipidemia groups were observed (table 9).
Table 9. Lipid peroxidation and antioxidant defense system biomarkers in normolipidemic and dyslipidemia groups.
|
Parameter |
Normolipidemic group |
Dyslipidemia group |
p-value |
|
N |
32 |
54 |
- |
|
MDA* (µmol/l)
|
1.58 (0.41) |
1.79 (0.60) |
3.946*10-2 |
|
TAC* (µmol/l)
|
386.97 (107.08) |
402.14 (116.38) |
0.911 |
|
SOD inhibition rate* (%)
|
75.85 (5.60) |
76.84 (9.23) |
0.514 |
*MDA – malondialdehyde, SOD inhibition rate – superoxide dismutase inhibiton rate, TAC – total antioxidant capacity, WC – waist circumference.
Regarding the menopausal status of study participants, we completely agree that physiological changes during menopause affect lipid metabolism and are important in assessing lipid metabolism biomarkers. This is especially important as our sample included women aged from 50 to 64 years old and menopausal status would greatly improve our study. Indeed, we included menopausal status in our study questionnaire, but the results could not be applied to the data analysis as the majority of study participants did not disclose this information or did not know their actual menopausal status: perimenopause, menopause or postmenopause. In order to clarify this, we added further statement to our publication:
Finally, there was no data available on menopausal status of study participants. It is well-established that menopause is associated with changes in lipid metabolism biomarkers such as lipoproteins, LDL-cholesterol, HDL-cholesterol and triacylglycerols [32]. Therefore, we would like to disclose that these changes were not addressed in this study, and we encourage other researchers to include the menopausal status of participants in similar studies.
Ko SH, Kim HS. Menopause-associated lipid metabolic disorders and foods beneficial for postmenopausal women. Nutrients 2020; 12(1): 202. doi:10.3390/nu12010202.
2.Levels of lipid biomarkers do not differ significantly between groups divided at cut-off accepted for MDA, except total cholesterol; for apoB there is only borderline difference. At cut-offs applied for TAC and SOD significant differences were found for apoB whereas for TG only at a cut-off for TAC.
Thank you for the remark. Our primary aim was to assess the differences and relationships between anthropometric measures and lipid metabolism biomarkers at MDA, TAC and SOD inhibition rate cut-off values. In the text we noted that ApoB had only statistically significant borderline difference: Interestingly, a borderline statistically significant result was observed in the comparison of ApoB levels between the groups: higher ApoB concentrations were observed in women group with higher MDA concentration.
3.Correlations between MDA, TAC and age are very weak; more interesting is a good significant correlation found between TAC and WC. Correlations between MDA, TAC and SOD with apoB and TG, though significant, are weak or very weak, except this of TG with TAC. Correlations do not say much and sometimes may be quite accidental.
Thank you for the valuable comment. In the results section we added the strength of relationships, so it would be easier to distinguish between weak and moderate correlations.
Correlation analysis revealed statistically significant although weak relationship between serum MDA values and age. Moreover, TAC had a statistically significant weak relationship with age, but moderate strength relationships with BMI and waist circumference. There wasere no statistically significant relationshiprelationships between serum MDA values and BMI or waist circumference. Finally, there were no statistically significant relationships between SOD inhibition rate and age, BMI, or waist circumference (Table 4).
Correlation analysis revealed that there were statistically significant weak relationships between MDA, TAC, as well as SOD inhibition rate levels and triacylglycerol concentration. Interestingly, TAC and triacylglycerol concentration had a statistically significant moderate strength relationship. In addition, MDA, TAC and SOD inhibition rate values had statistically significant but weak relationship with ApoB concentration. Furthermore, MDA and SOD inhibition rate values had statistically significant weak relationship with total-cholesterol concentration. There was statistically significant relationship between MDA and non-HDL-cholesterol values, although the strength of relationship was weak. Lastly, there were no other statistically significant relationships observed between lipid metabolism biomarkers and MDA, TAC or SOD inhibition rate (table 7).
4.Methods for measuring MDA, TAC and SOD are not standardized and not harmonized as the authors admit themselves; moreover these are time-consuming methods which could be hardly applied in clinical and laboratory practice. Evaluation of new potential biomarkers should clearly indicate their advantage when applied in the diagnosis. Finally, based on presented data, I do not see any additional value of measuring MDA, TAC and SOD as biomarkers of atherosclerosis risk.
Thank you for the comment. We decided to rewrite our conclusion so it would better suit our study and results:
In conclusion, the association of oxidative status indices, MDA, TAC and SOD inhibition rate, with cardiovascular risk factors suggests that they could be additional useful biomarkers in the research of aging, obesity, and atherosclerosis pathogenesis.
Minor comments:
1.The terms “apoproteins A or B” are wrongly used, instead apolipoproteins A or B should be used.
We corrected the terms throughout the text as suggested.
- Surprisingly, section 3- Results (lines 120-122) starts with something like instruction “how to prepare the manuscript”.
It was left by accident from the manuscript template. It was deleted.
Reviewer 3 Report
This is an interesting study. For the benefit of the reader, there are, however, still several questions need to be answered and clarified.
Major concern:
1. The lipid peroxidation biomarker malondialdehyde may be associated with some biomarkers. But the author should describe the physiological implications for cardiovascular disease prevention from this present study results more detail.
2. The subjects are healthy women, therefore, the authors should describe whether the results can fit healthy man.
3. English check of the manuscript should be performed by native English-speaking professional before re-submit the manuscript.
4. Some subjects have 43.44(BMI) and 127cm(waist circumference). Are they health?
5. Authors should explain why subjects with cardiovascular disease were not explored but healthy women.
Author Response
Dear Editor,
We thank the Reviewers for the valuable comments. They are given below in black fonts together with our answers to the comments:
The lipid peroxidation biomarker malondialdehyde may be associated with some biomarkers. But the author should describe the physiological implications for cardiovascular disease prevention from this present study results more detail.
Although the results of the current study showed statistically significant associations between malondialdehyde, antioxidant defense system components and lipid metabolism biomarkers, the application of these measures for the assessment of CVD risk is debatable. Currently, CVD risk estimation, individual-level interventions and treatment goals are based on the assessment of conventional risk factors including low physical activity, unbalanced diet, obesity, smoking, dyslipidemia, hypertension, and the presence of diabetes [1]. It is suggested that atherosclerosis and coronary artery calcification may be the result of oxidative stress and reactive oxygen species are involved in endothelial dysfunction, atheromatous plaque formation and rupture, measurement of circulating biomarkers of oxidative stress in serum or plasma is challenging because of the highly reactive nature of these molecules [1-3]. Moreover, there is no consensus which oxidative stress/antioxidant defense system measure (e.g., plasma total antioxidant status, F2-isoprostanes, lipid peroxides, thiobarbituric acid-reactive substances and monoaldehyde, biomarkers of protein peroxidation) is the most reliable “gold” standard method for the evaluation of oxidative damage in vivo [3]. Thus, larger prospective studies are needed to identify a universal and “easy to measure” marker of oxidative stress which might be involved in the current CVD prevention guidelines. We added this information in the Discussion section of the revised manuscript.
The subjects are healthy women, therefore, the authors should describe whether the results can fit healthy man.
Thank you for the remark. The results are not applicable to healthy man as there are well-known gender differences in oxidative stress. According to the literature, women are less susceptible to oxidative stress and there are clear differences in the expression and/or activities of antioxidant enzymes (catalase, superoxide dismutase, glutathione peroxidase) between males and females [4]. We added this information in the Discussion section of the revised manuscript.
English check of the manuscript should be performed by native English-speaking professional before re-submit the manuscript.
Thank you for the suggestion. The manuscript was carefully checked by native English-speaking professional.
Some subjects have 43.44(BMI) and 127cm(waist circumference). Are they health?
Thank you for the valuable comment. We totally agree that some of the subjects had higher than recommended body mass index or waist circumference. However, we use the term “apparently healthy” to emphasize that study subjects do not suffer from acute or chronic diseases, but they have different CVD risk factors including obesity, dyslipidemia, or hypertension.
Authors should explain why subjects with cardiovascular disease were not explored but healthy women.
Thank you for the comment. The objective of the study was to investigate the role of oxidative stress and antioxidant defense system components during the early development of atherosclerosis in subjects without clinically evident cardiovascular disease. Thus, we explored the association between major CVD risk factors (obesity, dyslipidemia, hypertension) and oxidant/antioxidant status in apparently healthy subjects. Moreover, the measurement of oxidative stress biomarkers in patients with cardiovascular disease is particularly challenging because the surgical procedures or medication treatment might influence oxidative stress status.
References
- Visseren FLJ, Mach F, Smulders YM, Carballo D, Koskinas KC, Bäck M, Benetos A, Biffi A, Boavida JM, Capodanno D, Cosyns B, Crawford C, Davos CH, Desormais I, Di Angelantonio E, Franco OH, Halvorsen S, Hobbs FDR, Hollander M, Jankowska EA, Michal M, Sacco S, Sattar N, Tokgozoglu L, Tonstad S, Tsioufis KP, van Dis I, van Gelder IC, Wanner C, Williams B; ESC Scientific Document Group. 2021 ESC Guidelines on cardiovascular disease prevention in clinical practice. Eur J Prev Cardiol. 2022;29(1):5-115. doi: 10.1093/eurjpc/zwab154.
- Lee R, Margaritis M, Channon KM, Antoniades C. Evaluating oxidative stress in human cardiovascular disease: methodological aspects and considerations. Curr Med Chem. 2012;19(16):2504-20. doi: 10.2174/092986712800493057.
- Stephens JW, Khanolkar MP, Bain SC. The biological relevance and measurement of plasma markers of oxidative stress in diabetes and cardiovascular disease. Atherosclerosis. 2009;202(2):321-9. doi: 10.1016/j.atherosclerosis.2008.06.006.
- Kander MC, Cui Y, Liu Z. Gender difference in oxidative stress: a new look at the mechanisms for cardiovascular diseases. J Cell Mol Med. 2017;21(5):1024-1032. doi: 10.1111/jcmm.13038.
Round 2
Reviewer 1 Report
Revised manuscript has improved.
Reviewer 2 Report
The article has been sufficiently improved. All my suggestions to change were included (for. ex. additional tables). Discussion and conclusions were sufficiently modified.
Minor corrections required.
Reviewer 3 Report
The manuscript is now suitable for publication in this journal.